# Plasma eNOS Concentration in Healthy Pregnancy and in Hypertensive Disorders of Pregnancy: Evidence of Reduced Concentrations in Pre-Eclampsia from Two Independent Studies

**DOI:** 10.3390/diseases11040155

**Published:** 2023-11-01

**Authors:** Julyane N. S. Kaihara, Caroline K. Minami, Maria T. S. Peraçoli, Mariana Romão-Veiga, Vanessa R. Ribeiro-Vasques, José C. Peraçoli, Ana C. T. Palei, Ricardo C. Cavalli, Priscila R. Nunes, Marcelo R. Luizon, Valeria C. Sandrim

**Affiliations:** 1Department of Biophysics and Pharmacology, Institute of Biosciences of Botucatu, Sao Paulo State University (UNESP), Botucatu 18618-689, SP, Brazil; j.kaihara@unesp.br (J.N.S.K.); c.minami@unesp.br (C.K.M.); priscila.nunes@unesp.br (P.R.N.); mrluizon@ufmg.br (M.R.L.); 2Department of Gynecology and Obstetrics, Botucatu Medical School, Sao Paulo State University (UNESP), Botucatu 18618-687, SP, Brazil; terezinha.peracoli@unesp.br (M.T.S.P.); romao.veiga@unesp.br (M.R.-V.); jc.peracoli@unesp.br (J.C.P.); 3Department of Chemistry and Biological Sciences, Institute of Biosciences of Botucatu, Sao Paulo State University (UNESP), Botucatu 18618-689, SP, Brazil; vanessa.ribeiro@unesp.br; 4Department of Surgery, School of Medicine, University of Mississippi Medical Center, Jackson, MS 39216, USA; apalei@umc.edu; 5Department of Gynecology and Obstetrics, Faculty of Medicine of Ribeirao Preto, University of Sao Paulo (USP), Ribeirao Preto 14049-900, SP, Brazil; rcavalli@fmrp.usp.br; 6Department of Genetics, Ecology and Evolution, Institute of Biological Sciences, Federal University of Minas Gerais, Belo Horizonte 31270-901, MG, Brazil

**Keywords:** endothelial nitric oxide synthase, gestational hypertension, nitric oxide, NOS3 protein, pre-eclampsia, pregnancy

## Abstract

Hypertensive disorders of pregnancy (HDP), comprising gestational hypertension (GH) and pre-eclampsia (PE), are leading causes of maternal and perinatal morbidity and mortality. Both GH and PE are characterized by new-onset hypertension, but PE additionally includes proteinuria and/or end-organ damage. Impaired nitric oxide (NO) bioavailability may lead to endothelial dysfunction in GH and PE, and the primary source of vascular NO is endothelial NO synthase (eNOS). However, no previous study has investigated plasma eNOS concentrations in patients with GH and PE. In this study, we compared plasma eNOS concentrations in healthy pregnancies and HDP in two independent cohorts. The primary study included 417 subjects, with 43 non-pregnant (NP) and 156 healthy pregnant (HP) women and 122 patients with GH and 96 with PE. The replication study included 85 pregnant women (41 healthy and 44 pre-eclamptic). Plasma concentrations of eNOS were measured using a commercial ELISA kit provided by R&D Systems, and plasma nitrite concentrations were assessed using two ozone-based chemiluminescence assays. Correlations between plasma eNOS concentrations and plasma nitrite concentrations, as well as clinical and biochemical parameters, were evaluated by either Spearman’s or Pearson’s tests. In the primary study, NP women and HDP had significantly lower plasma eNOS concentrations compared to HP; concentrations were even lower in PE compared to GH. Plasma eNOS concentrations were reduced but not significant in early-onset PE, PE with severe features, preterm birth, and intrauterine growth restriction. No correlation was observed between plasma eNOS and nitrite levels. In HDP, there was a significant positive correlation between levels of eNOS and hemoglobin (*r* = 0.1496, *p* = 0.0336) as well as newborn weight (*r* = 0.1487, *p* = 0.0316). Conversely, a negative correlation between eNOS levels and proteinuria was observed (*r* = −0.2167, *p* = 0.0179). The replication study confirmed significantly reduced plasma concentrations of eNOS in PE compared to HP. Our findings provide evidence of reduced plasma eNOS concentrations in HDP; they were particularly lower in PE compared to GH and HP in two independent studies.

## 1. Introduction

Pre-eclampsia (PE), a specific syndrome of human gestation, affects 2 to 8% of all pregnancies worldwide [1], and its epidemiological importance is due to the fact that this disease is considered one of the main causes of maternal death. In Brazil, PE is responsible for 37% of obstetric causes of death [2].

The multifactorial nature of the disease characterizes this condition as heterogeneous and diverse because PE is currently recognized as a heterogeneous syndrome. Initially, PE may be characterized by new-onset arterial hypertension and also proteinuria from the twentieth week of gestation or the first days after delivery. More recently, other sets of symptoms have also aggregated the classification of the disease in the absence of proteinuria. Maternal dysfunctions that may affect pregnancy are also related to PE, such as neurological or hematological complications, kidney failure, liver involvement, placental dysfunction, and fetal growth restriction [1].

Several pathways have been proposed for PE development. The generalized endothelial dysfunction, characteristic of this disease, is responsible for the clinical abnormalities, altering levels of vasoactive substances, such as nitric oxide (NO), prostacyclin, and endothelin [3,4,5,6,7].

NO is an essential gaseous molecule produced by the conversion of L-arginine to L-citrulline, which is catalyzed by NO synthases [8]. NO acts as a potent endogenous regulator of vascular homeostasis, with an important role in blood pressure maintenance, and it is involved in other physiological cellular functions, such as immune response and proliferation [9]. Since NO inhibits the calcium-induced influx to the smooth muscle cell, NO has a central role in vasodilation via soluble guanylyl cyclase [10]. Therefore, any imbalance in the bioavailability of NO can lead to endothelial dysfunction, which is a common feature in most hypertensive disorders [9,11,12,13], including PE, as mentioned.

Hypertensive disorders of pregnancy (HDP) are leading causes of perinatal and maternal morbidity and mortality, and they are marked by increased blood pressure during gestation [14,15]. Gestational hypertension (GH) and pre-eclampsia are defined by new-onset hypertension after 20 weeks of gestation, but only PE shows proteinuria and/or end-organ damage [1]. Both GH and PE are associated with complications in multiple organ systems: circulatory, cardiac, hepatic, and central nervous system complications, and eclampsia [14]. It is well-known that impaired NO bioavailability is usually present in HDP [16,17], and it is linked to endothelial dysfunction in PE [7,18]. Therefore, it is reasonable to assume that the pathway of NO production is impaired in GH and PE.

Endothelial nitric oxide synthase (eNOS, encoded by the *NOS3* gene) is one of the three isoforms of NOS enzymes and the primary source of NO production in the vasculature. Since eNOS is the constitutive isoform of endothelial cells and its activity is directly related to vascular dilation [19], its decreased activity and consequent diminished NO levels could be implicated in several endothelial disorders and contribute to HDP. Interestingly, eNOS may be carried out by syncytiotrophoblasts in maternal circulation by binding extracellular vesicles, which could explain the presence of eNOS in the plasma [20]. This study also observed lower activity in syncytiotrophoblast extracellular vesicle-bound eNOS in women with PE, possibly contributing to the lower NO levels in patients with PE [20]. Considering the association between NO derived from eNOS and the development of endothelial dysfunction as a hallmark of hypertension, we hypothesized that plasma eNOS concentrations would be reduced in HDP compared to healthy pregnancy.

In this study, we measured eNOS concentrations in plasma from non-pregnant and healthy pregnant women and in patients with HDP and correlated them with clinical parameters in a primary study. Next, we further measured plasma eNOS concentrations in healthy pregnant women and in patients with PE in a replication study.

## 2. Materials and Methods

### 2.1. Subjects

The primary study was approved by the Institutional Review Board at Ribeirao Preto Medical School of University of Sao Paulo (FMRP-USP; 37738620.0.0000.5440, 19 October 2020), and enrolled 417 women at the Department of Obstetrics and Gynecology, University Hospital at FMRP-USP. The women were non-pregnant (NP, *n* = 43) or healthy pregnant (HP, *n* = 156) and had hypertensive disorders of pregnancy (HDP, *n* = 218), defined as gestational hypertension (GH, *n* = 122) and PE (*n* = 96) according to ACOG criteria [1]. The criteria of PE included new-onset hypertension (at least 140 mmHg and/or 90 mmHg) after 20 weeks of gestation associated with proteinuria (≥300 mg/24 h), or, in the absence of this, thrombocytopenia (platelet count < 100 × 10^9^/L), renal insufficiency (serum creatinine concentration > 1.1 mg/dL or a doubling of the serum creatinine concentration in the absence of other renal diseases), impaired liver function, pulmonary edema, and brain or visual symptoms. PE subtypes were defined according to ISSHP [21]: patients with pre-eclampsia with severe features presenting blood pressure > 160 mmHg systolic or 110 mmHg diastolic; early-onset pre-eclampsia patients with the onset of symptoms occurring before 34 weeks of pregnancy; preterm pre-eclampsia occurring between 34^+1^ and 37^+0^ weeks of pregnancy; and intrauterine growth restriction (IUGR) defined as an estimated fetal weight below the 10th percentile for gestational age [22]. GH was defined as new-onset hypertension (at least 140 mmHg and/or 90 mmHg) after 20 weeks of gestation without proteinuria and/or the aforementioned symptoms of end-organ damage [1]. The replication study was approved by the Research Ethics Committee at Botucatu School of Medicine of Sao Paulo State University (4.961.945, 9 September 2021) and included 85 women: healthy pregnant (HP, *n* = 41) and PE (*n* = 44).

Both studies were performed in accordance with the Declaration of Helsinki. Exclusion criteria for all groups were: hemostatic abnormalities, chronic hypertension, cancer, multiple pregnancy, diabetes, and cardiovascular, autoimmune, renal, and hepatic diseases. At clinical attendance, written informed consent was obtained from all women involved in the study. For pregnant women younger than 18 years old, the written informed consent was obtained from their parents or guardians. Demographic and clinical data were gathered from medical records from results of the routine laboratory tests conducted on pregnant women and their respective newborns. The blood of pre-eclamptic women was obtained at the time of diagnosis, while that of normotensive pregnant women was obtained at the time of their attendance. Blood samples (10 mL) were collected by venipuncture from the antecubital vein and were put into a sterile plastic tube containing 10 U/mL EDTA (Becton Dickinson-BD Vacutainer; BD Biosciences, Franklin Lakes, NJ, USA). After centrifugation for 10 min at 1000× *g*, the obtained plasma was removed, and aliquots were stored at −80 °C until the time of eNOS determination.

### 2.2. Measurement of eNOS Concentrations

Plasma eNOS concentrations were measured by enzyme-linked immunosorbent assay (ELISA) using the commercially available DuoSet ELISA kit for Human eNOS (catalog #DY950-05; R&D Systems, Minneapolis, MN, USA), according to manufacturer’s instructions. In the test, the concentrations of capture and detection antibodies, as well as specific recombinant human eNOS standards used in standard curves, were those recommended by the manufacturer (R&D Systems). Briefly, 96-well microplates (cat. #675061, Greiner Bio-One, Frickenhausen, Germany) were incubated with capture antibody overnight, followed by washes using Tween^®^ 20/PBS (phosphate-buffered saline; 137 mM NaCl, 2.7 mM KCl, 8.1 mM Na_2_HPO_4_, 1.5 mM KH_2_PO_4_, and pH 7.2–7.4) solution. Next, plates were blocked using bovine serum albumin (BSA, cat. #A-8022; Sigma-Aldrich, St. Louis, MO, USA), diluted to 1% concentration with PBS, and washed again. Each plasma sample was diluted with PBS at a 1:100 ratio. Plates were incubated with the standard protein and samples, washed, and incubated with the detection antibody. After a further washing step, plates were incubated with Streptavidin conjugated to horseradish-peroxidase (Streptavidin-HRP) and subsequently incubated with a 3,3′,5,5′-Tetramethylbenzidine Liquid Substrate System for ELISA (#T0440; Sigma-Aldrich, St. Louis, MO, USA), both in the dark. Then, 2 N H_2_SO_4_ solution was added to stop the reaction. All incubations occurred for the required duration for each step, as instructed by the manufacturer, at room temperature. The absorbance was measured at a wavelength of 450 nm using a multi-plate reader (Synergy 4, Biotek Instruments, Winooski, VT, USA) to obtain the optical density. Concentrations and optical densities were logarithmically linearized, and regression analysis was performed on the data. The concentrations of eNOS were calculated in pg/mL from the standard curve generated with the respective human recombinant standard.

### 2.3. Measurement of Nitrite Concentrations

The methodologies used to measure plasma nitrite concentrations were ozone-based chemiluminescence assays, as previously described [16,17]. For selected samples of the primary study, nitrite levels were assessed in triplicate through the injection of 200 μL of plasma samples into a tri-iodide solution that had been acidified and cleansed with nitrogen, in conjunction with the use of a gas-phase chemiluminescence NO analyzer (Sievers Model 280 NO Analyzer, General Electric Company, Boulder, CO, USA). To the purge vessel, approximately 8 mL of tri-iodide solution containing 2.0 g of potassium iodide, 1.3 g of iodine solubilized in 40 mL of water, and 140 mL of acetic acid was added. For the ascorbic acid reduction assay, a volume of 100 μL of plasma was injected into a solution consisting of 8 mL of glacial acetic acid and 60 mM of ascorbic acid. This solution was then purged with nitrogen in line with gas-phase chemiluminescence in the NO analyzer in the system described previously. Both the tri-iodide and ascorbic acid solutions reduce nitrites to NO gas, which was subsequently measured by the NO analyzer.

### 2.4. Statistical Analysis

Demographic, clinical, and biochemical parameters were evaluated by normality tests, and differences between study groups were analyzed using Student *t*-test (Mann–Whitney test) or One-Way ANOVA (Kruskal–Wallis test and Dunn’s post-test), as appropriate. Plasma eNOS data were reported as the logarithm of its concentration. Relationships between eNOS levels and plasma nitrite concentrations, as well as clinical and biochemical parameters, were analyzed using Spearman’s or Pearson’s correlation (*r* and *p*-values), as appropriate. Analysis was performed using GraphPad Prism (GraphPad Software Inc., San Diego, CA, USA). A value of *p* < 0.05 was considered the level of statistical significance.

## 3. Results

### 3.1. Primary Study: Demographic and Clinical Characteristics and Plasma eNOS Concentrations

The characteristics of the 417 women included in the primary study are shown in Table 1. HP and HDP women were younger than the NP women (*p* < 0.0001). BMI, SBP and DBP were higher in HDP women compared to other groups (all *p* < 0.0001). HDP women showed lower newborn weight and gestational age at delivery than HP women (both *p* < 0.001). The gestational age at delivery and newborn weight were significantly lower in the HDP compared to the HP group (37.8 ± 3.0 vs. 39.6 ± 1.4, and 2992.0 ± 765.3 vs. 3305.0 ± 525.2, respectively), as expected. Conversely, there were no significant differences between the groups in terms of ethnicity, smoking status, heart rate, platelet count, urea levels, or liver enzymes (serum glutamic-oxaloacetic transaminase, SGOT) (all *p* > 0.05).

In the primary study, plasma eNOS concentrations were significantly reduced both in NP women and HDP women, compared to the HP group (*p* = 0.0001; Figure 1a).

We then analyzed patients with HDP by dividing into subgroups of women with GH and PE (the demographic and clinical characteristics for the groups of women with GH and PE are shown in Table 2). PE women showed higher SBP and urea levels (both *p* < 0.01). Conversely, gestational age at delivery and newborn weight were lower in PE than in GH women (both *p* ≤ 0.001). As expected, proteinuria was higher in PE than in GH women (*p* < 0.001). Plasma eNOS concentrations were significantly lower in PE than in GH women (*p* = 0.0137; Figure 1b). A comparison between HP and GH showed no significant difference (*p* = 0.0708; Appendix A), and plasma eNOS concentrations were lower in NP compared to in GH women (*p* = 0.0123; Appendix A).

We further examined whether plasma eNOS concentrations differ between the subgroups of PE women, including early-onset versus late-onset PE, severity of PE, preterm birth, and intrauterine growth restriction (IUGR) (Appendix A). Interestingly, subgroups of early-onset PE, PE with severe features, preterm birth, and IUGR showed lower plasma eNOS concentrations, but these results were not significant (*p* > 0.05).

### 3.2. Replication Study: Clinical and Biochemical Characteristics and Plasma eNOS Concentrations

The characteristics of the 85 women included in the replication study are shown in Appendix A. Gestational age at sampling was higher in PE patients than in HP patients (*p* = 0.030). We found that plasma eNOS concentrations were significantly lower in PE than in HP patients (*p* < 0.05; Figure 1c), and the results were similar to those found in the primary study.

### 3.3. Correlations between eNOS Concentrations, Nitrite Concentrations, and Clinical and Biochemical Characteristics in the Primary Study

Finally, we analyzed the relationship between plasma eNOS concentrations and nitrite concentrations (Appendix A), but no significant correlations were found among the groups HDP (*r* = −0.1290, *p* = 0.1567), PE (*r* = −0.0906, *p* = 0.4875), GH (*r* = −0.1180, *p* = 0.3908), and HP (*r* = −0.0952, *p* = 0.4264). Correlations between plasma eNOS concentrations and clinical and biochemical characteristics of subjects in the primary study are shown in Table 3. In HDP women, we found positive correlations between plasma eNOS concentrations and hemoglobin (*r* = 0.1496, *p* = 0.0336) and newborn weight (*r* = 0.1487, *p* = 0.0316). Conversely, we found negative correlations between plasma eNOS concentrations and proteinuria in HDP women (*r* = −0.2167, *p* = 0.0179). However, we found no significant correlations between plasma eNOS concentrations and clinical or biochemical parameters within HP, GH, and PE patients (*p* > 0.05; Table 3).

## 4. Discussion

This study was the first to measure the plasmatic eNOS concentrations in patients with HDP and to correlate them with clinical parameters. The main novel findings reported here are that: in the primary study, (1) eNOS concentrations were reduced in plasma from patients with HDP compared to HP, and (2) were even lower in PE than GH; (3) eNOS concentrations were positively correlated with hemoglobin and newborn weight and inversely correlated with proteinuria in HDP in the primary study. In the replication study, we confirmed that (4) plasma eNOS concentrations were lower in PE than HP. These data may contribute to understanding the role of endothelial dysfunction in HDP.

Importantly, there is a close relationship between placental and circulating levels of eNOS. The presence of vesicles responsible for carrying eNOS from the placenta to the maternal circulation was previously described and shows the reduced concentration and activity of eNOS [20]. Additionally, a reduced expression of eNOS was found in endothelial cells from PE patients due to the upregulation of placental vesicles [23]. Moreover, circulating extracellular vesicles from pregnant women with PE with severe features significantly downregulated eNOS and phospho-eNOS proteins when compared to extracellular vesicles from normotensive pregnant women [24].

Conversely, two previous studies from the same group found no differences for serum eNOS levels between PE with severe features and healthy pregnancy [25,26]. The use of different ELISA kits and the use of serum instead of plasma may explain the contrasting results compared to those of the present study. In addition, the eNOS data in these previous studies were reported using U/m—a unit of measurement for enzyme activity. As the group did not indicate the catalog number of the assay, the unit used in the only eNOS ELISA kit available in the company (Wuhan USCN) was ng/mL. Thus, we are not sure how to compare these data with ours.

Gene regulation may explain differences in eNOS levels, and there are mechanisms of post-transcriptional modulation using miRNAs [27]. Specific miRNAs directly affecting eNOS expression may be induced in different conditions and affect multiple targets. Another context to explore is the correlation between the expression of placental eNOS and increased oxidative stress in women with PE [28], which could cause BH4 (tetrahydrobiopterin) oxidation and eNOS uncoupling [27]. In this context, miRNAs could be relevant, as several miRNAs may affect placental eNOS expression and are regulated by the consumption of antioxidants [29]. In the present study, both scenarios are plausible, and a promising strategy could be to increase eNOS levels and ameliorate the endothelial dysfunction observed in hypertensive disorders of pregnancy.

The main consequence of reduced eNOS levels could be impaired NO production, which is critical for PE and the development of HDP. During pregnancy, if impaired NO production creates an environment susceptible to endothelial dysfunction, it could affect feto-placental blood flow and embryonic development. The placenta depends on the NO balance to provide vascular adaptation and maintenance [30], which is the reason it could be a leading point for studies on HDP. We are aware that NO bioavailability relies not only on eNOS expression or activity but also on the presence of cofactors, substrates, and enzyme integrity [27]. eNOS levels have been explored in vascular disorders [31,32]. Reduced plasma eNOS levels were found in slow coronary flow and were related to endothelial dysfunction [32], which is consistent with the present findings of reduced eNOS levels in HDP.

No correlations were found among circulating eNOS and nitrite concentrations in any group. It is difficult to observe an association between these two biomarkers because mainly nitrite (the NO bioavailability biomarker) may suffer from the interference of many factors, such as the presence of superoxide, which scavenges NO, not necessarily representing a direct effect of eNOS activity. Moreover, we measured eNOS levels and not their activity, which may be better correlated to nitrite levels.

To evaluate the impact on clinical outcomes, we observed a positive correlation between plasma eNOS levels and hemoglobin levels in patients with HDP. Notably, red blood cells were previously shown to have their own contribution to circulatory NO bioavailability [33], so it is consistent to admit that a low rate of hemoglobin is associated with reduced levels of circulating eNOS. Moreover, assessing how whole-blood nitrite concentrations reflect constitutive NOS activity [34,35] is of key relevance to studying endogenous NO production [36]. We observed a significant difference in newborn weight between HP and HDP and between GH and PE patients. This may be related to the gestational age at delivery of our subjects, which also presented a significant difference, as previous studies showed that PE has a strong association with low birth weight and found that decreased newborn weight is a function of gestational age of PE and GH pregnancies [37,38,39]. The correlation between plasma eNOS levels and newborn weight may occur due to that. The negative correlation found between plasma eNOS concentration and proteinuria was expected because high proteinuria is a diagnostic criterion for PE, and eNOS concentrations were lower in HDP.

The present study has limitations. First, we only evaluated plasma eNOS levels from women in the third trimester of gestation, and it is important to analyze these parameters at other gestational ages. Furthermore, it is possible that even with reduced plasma eNOS concentrations, we would find no changes in the eNOS activity of PE or GH. Further studies are warranted to elucidate both eNOS levels and activity in HDP, mainly in PE.

## 5. Conclusions

We provided evidence for reduced plasma eNOS concentrations in HDP, which were particularly lower in patients with PE than in healthy pregnancies, from two independent studies. Our novel findings may help us to understand the role of eNOS in the pathogenesis of PE. Furthermore, plasma eNOS levels might also be further explored to help in the early prediction of the HDP, mainly in PE.

## Figures and Tables

**Figure 1 diseases-11-00155-f001:**
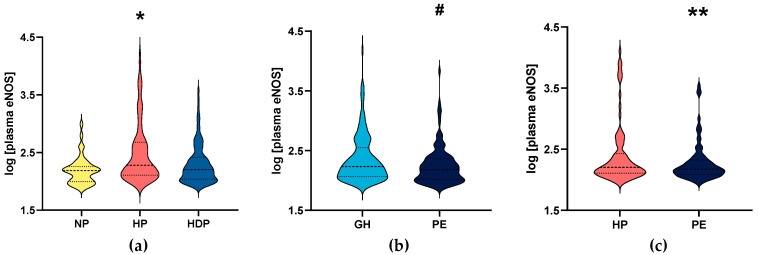
Logarithm of plasma endothelial nitric oxide synthase (eNOS) concentrations from subjects enrolled in the primary study: (**a**) non-pregnant (NP) and healthy pregnant (HP) women, and hypertensive disorders of pregnancy (HDP), including (**b**) gestational hypertension (GH) and pre-eclampsia (PE) patients; and (**c**) healthy pregnant women (HP) and patients with pre-eclampsia (PE) enrolled in the replication study. * (*p* < 0.05) when compared to NP and HDP. # (*p* < 0.05) when compared to GH. ** (*p* < 0.05) when compared to healthy pregnant women (HP). Median values are represented by dashed lines and the first and third quartiles are shown as dotted lines.

**Table 1 diseases-11-00155-t001:** Clinical, demographic, and biochemical characteristics of subjects enrolled in the primary study.

Parameters	Non-Pregnant	Healthy Pregnant	Hypertensive Disorders of Pregnancy	*p*-Value
	(*n* = 43)	(*n* = 156)	(*n* = 218)	
Age (years)	32.4 ± 10.2	24.8 ± 6.0	27.5 ± 6.6	**<0.0001**
Ethnicity (% white)	79.2	65.4	66.4	0.0548
Current smokers (%)	6.1	12.6	13.0	0.2129
BMI (kg/m^2^)	22.7 ± 3.0	28.2 ± 4.8	35.0 ± 6.7	**<0.0001**
SBP (mmHg)	115.6 ± 8.4	110.9 ± 11.2	134.3 ± 18.0	**<0.0001**
DBP (mmHg)	78.1 ± 5.3	71.6 ± 8.7	84.0 ± 12.5	**<0.0001**
HR (beats/min)	ND	81.8 ± 9.5	81.8 ± 7.3	0.7697
Platelets (×10^3^/mm^3^)	ND	225.2 ± 71.4	227.1 ± 61.6	0.5433
Urea levels (mg/dL)	ND	15.8 ± 5.2	17.0 ± 14.2	0.9374
SGOT (U/L)	ND	21.2 ± 7.1	18.6 ± 16.8	0.0734
24 h Pr (mg/24 h)	ND	ND	861.4 ± 1680.0	–
GAD (weeks)	NA	39.6 ± 1.4	37.8 ± 3.0	**<0.0001**
Newborn weight (g)	NA	3305.0 ± 525.2	2992.0 ± 765.3	**0.0006**

Abbreviations: BMI, body mass index; SBP, systolic blood pressure; DBP, diastolic blood pressure; HR, heart rate; SGOT, serum glutamic-oxaloacetic transaminase; 24 h Pr, 24 h proteinuria; GAD, gestational age at delivery; ND, not determined; NA, not applicable. Data are expressed as mean ± S.D. or percentage. Bold values are significant *p*-values.

**Table 2 diseases-11-00155-t002:** Clinical, demographic, and biochemical characteristics of patients with gestational hypertension and pre-eclampsia in the primary study.

Parameters	Gestational Hypertension	Pre-Eclampsia	*p*-Value
	(*n* = 122)	(*n* = 96)	
Age (years)	27.5 ± 6.8	27.5 ± 6.4	0.8719
Ethnicity (% white)	70.2	61.5	0.1807
Current smokers (%)	14.3	11.7	0.6741
BMI (kg/m^2^)	35.7 ± 7.0	34.1 ± 6.3	0.2130
SBP (mmHg)	131.9 ± 17.9	137.3 ± 17.8	**0.0065**
DBP (mmHg)	83.2 ± 12.2	85.1 ± 12.8	0.1703
HR (beats/min)	81.9 ± 7.5	81.7 ± 7.0	0.9486
Platelets (×10^3^/mm^3^)	232.5 ± 59.1	220.5 ± 64.2	0.2405
Urea levels (mg/dL)	14.4 ± 5.5	20.3 ± 19.9	**0.0004**
SGOT (U/L)	18.8 ± 21.1	18.5 ± 9.5	0.7236
24 h Pr (mg/24 h)	168.7 ± 79.3	1457 ± 2121	**<0.0001**
GAD (weeks)	39.0 ± 1.6	36.5 ± 3.6	**<0.0001**
Newborn weight (g)	3215 ± 524.7	2701 ± 919.6	**<0.0001**
Early-onset PE (%)	NA	25.6	–
Preterm birth (%)	ND	32.6	–
IUGR (%)	ND	29.8	–
Severity (%)	NA	39.5	–

Abbreviations: BMI, body mass index; SBP, systolic blood pressure; DBP, diastolic blood pressure; HR, heart rate; SGOT, serum glutamic-oxaloacetic transaminase; 24 h Pr, 24 h proteinuria; GAD, gestational age at delivery; IUGR, intrauterine growth restriction; ND, not determined; NA, not applicable. Data are expressed as mean ± S.D. or percentage. Bold values are significant *p*-values.

**Table 3 diseases-11-00155-t003:** Correlations between plasma eNOS concentrations and clinical/biochemical parameters in the primary study.

Clinical and Biochemical Parameters	Healthy Pregnant	Hypertensive Disorders of Pregnancy	Gestational Hypertension	Pre-Eclampsia
BMI (kg/m^2^)	−0.0072	−0.0002	−0.0387	0.0407
SBP (mmHg)	−0.0228	−0.0877	−0.0078	−0.1371
DBP (mmHg)	−0.0099	−0.0604	−0.0330	−0.0656
HR (beats/min)	0.1184	0.0125	0.0109	0.0077
Hemoglobin (g/dL)	−0.0472	**0.1496**	0.0823	0.1833
Hematocrit (%)	−0.1620	0.1227	0.0498	0.1909
Platelets (×10^3^/mm^3^)	0.2466	0.0995	0.1174	0.0363
Urea levels (mg/dL)	0.4308	0.1127	0.1669	0.1468
SGOT (U/L)	−0.3649	0.0965	0.0897	0.1233
24 h Pr (mg/24 h)	–	**−0.2167**	−0.0369	−0.1699
GAD (weeks)	0.0071	0.0531	−0.0923	0.0269
Newborn weight (g)	0.0113	**0.1487**	0.0267	0.1312

Abbreviations: BMI, body mass index; SBP, systolic blood pressure; DBP, diastolic blood pressure; HR, heart rate; SGOT, serum glutamic-oxaloacetic transaminase; 24 h Pr, 24 h proteinuria, GAD, gestational age at delivery. Values are the *r* Spearman or Pearson. Bold values are significant *p*-values.

## Data Availability

The data presented in this study are available on request from the corresponding author. The data are not publicly available due to privacy restrictions.

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
