# Peer review of "Plasma eNOS Concentration in Healthy Pregnancy and in Hypertensive Disorders of Pregnancy: Evidence of Reduced Concentrations in Pre-Eclampsia from Two Independent Studies"

_diseases, 2023, doi:10.3390/diseases11040155_

Round 1

Reviewer 1 Report

Comments and Suggestions for Authors

This is a well written and interesting paper with a well thought and executed experimental plan. The results showed 

1. plasma eNOS lower in non-pregnant (NP), hypertensive disorders of pregnancy (HDP) women compared to healthy pregnant women (HP).

2. eNOS lower in preeclampsia (PE) compared to gestational hypertensives (GH)

3. In HDP, eNOS levels were positively correlated with hemoglobin and newborn weight and inversely correlated with proteinuria

4. The replication study confirmed reduced plasma concentrations of eNOS in PE compared to HP (p < 0.05).

Plasma eNOS concentrations correlated positively with hemoglobin and with newborn weight, while negative correlations were found between plasma eNOS concentrations and proteinuria (only in HDP women).

Comments on the Quality of English Language

Mostly good. Some minor corrections required in places, e.g. replace "unbalance" with "imbalance" in line 69.

The word "expression" is normally reserved for RNA; authors should therefore use the word "levels" or "concentration" in plasma.

Author Response

Journal: Diseases (ISSN: 2079-9721)

Manuscript ID: diseases-2617240

Type: Article

Title: Plasma eNOS concentration in healthy pregnancy and in hypertensive disorders of pregnancy: evidence of reduced concentrations in preeclampsia from two independent studies

Julyane N. S. Kaihara, Caroline K. Minami, Maria T. S. Peraçoli, Mariana Romão-Veiga, Vanessa R. Ribeiro-Vasques, José C Peraçoli, Ana C. T. Palei, Ricardo C. Cavalli, Priscila R. Nunes, Marcelo R. Luizon, Valeria C. Sandrim

We thank the reviewers for their helpful comments and suggestions, and we have thoroughly revised our manuscript accordingly (revisions are highlighted in the manuscript file). Below, we present our responses to the reviewers, in bold.

Reviewer #1:

Comments and Suggestions for Authors

This is a well written and interesting paper with a well thought and executed experimental plan. The results showed: 

  1. plasma eNOS lower in non-pregnant (NP), hypertensive disorders of pregnancy (HDP) women compared to healthy pregnant women (HP).
  2. eNOS lower in preeclampsia (PE) compared to gestational hypertensives (GH)
  3. In HDP, eNOS levels were positively correlated with hemoglobin and newborn weight and inversely correlated with proteinuria
  4. The replication study confirmed reduced plasma concentrations of eNOS in PE compared to HP (p < 0.05).

Plasma eNOS concentrations correlated positively with hemoglobin and with newborn weight, while negative correlations were found between plasma eNOS concentrations and proteinuria (only in HDP women).

Comments on the Quality of English Language

Mostly good. Some minor corrections required in places, e.g. replace "unbalance" with "imbalance" in line 69.

We have corrected these mistakes.

The word "expression" is normally reserved for RNA; authors should therefore use the word "levels" or "concentration" in plasma.

We have corrected the identified errors (lines 293 and 234). In lines 258 and 261, eNOS expression refers to the enzyme expression in tissues, so we emphasized that using “expression of placental eNOS”.

Reviewer 2 Report

Comments and Suggestions for Authors

In the current study, the authors have examined the plasma eNOS concentrations in patients with hypertensive disorders during pregnancy to correlate with clinical parameters. According to authors, the main novel findings include in the primary study, (1) When compared to healthy pregnancy, the plasma of patients with hypertensive disorders had lower eNOS concentrations, and (2) eNOS conc were even lower in Preclampsia than gestational hypertension; (3) In hypertensive disorders, the eNOS concentrations were inversely connected with proteinuria and positively correlated with neonatal weight and hemoglobin. Also, authors have mentioned that impaired NO bioavailability is usually present in hypertensive disorders and it is linked to endothelial dysfunction in Preclampsia. And so they have hypothesized that the pathway of NO production is impaired in GH and PE. The authors should further check the eNOS specific NO as NO also comes from iNOS and antihypertensive effect has already been reported upon iNOS inhibition (PMID: 23890248)

General concept comments

1.     What was the NO level in these groups?

2.     What are levels of Arginine and ADMA as these are substrate and inhibitors of eNOS. Could you draw a correlation with the levels of Arginine, ADMA, eNOS, NO and clinical parameters in these groups. This would give us a more specific understanding of NOS pathway and would be more informative.

3.     How will you rule out the possibility of NO coming from iNOS

Specific comments

1.     How have you distinguished between PE and GH as both are HPD. Write in text in Abstract and Introduction for the clarity of readers.

2.     Fig 1: Please show the eNOS level comparison with GH and HP with statistical analysis and GH and NP with statistical analysis (as GH have a range of above 400 pg/ml while NP is below 300pg/ml, please check if there’s a significant difference)

3.     The author should also make a figure 2 showing plasma eNOS concentration in replicative study.

Comments on the Quality of English Language

Minor editing is needed

Author Response

Reviewer #2:

Comments and Suggestions for Authors

In the current study, the authors have examined the plasma eNOS concentrations in patients with hypertensive disorders during pregnancy to correlate with clinical parameters. According to authors, the main novel findings include in the primary study, (1) When compared to healthy pregnancy, the plasma of patients with hypertensive disorders had lower eNOS concentrations, and (2) eNOS conc were even lower in Preeclampsia than gestational hypertension; (3) In hypertensive disorders, the eNOS concentrations were inversely connected with proteinuria and positively correlated with neonatal weight and hemoglobin. Also, authors have mentioned that impaired NO bioavailability is usually present in hypertensive disorders and it is linked to endothelial dysfunction in Preeclampsia. And so they have hypothesized that the pathway of NO production is impaired in GH and PE. The authors should further check the eNOS specific NO as NO also comes from iNOS and antihypertensive effect has already been reported upon iNOS inhibition (PMID: 23890248)

General concept comments

  1. What was the NO level in these groups?
  2. What are levels of Arginine and ADMA as these are substrate and inhibitors of eNOS. Could you draw a correlation with the levels of Arginine, ADMA, eNOS, NO and clinical parameters in these groups. This would give us a more specific understanding of NOS pathway and would be more informative.
  3. How will you rule out the possibility of NO coming from iNOS

These points are noteworthy, but we have not quantified the levels of arginine, or ADMA in these samples. Regarding samples of population 2 (replication), plasma were not collected in stabilizer substance to conserve nitrite. Furthermore, the aim of this study solely involved quantifying the plasma eNOS concentrations and not exploring the enzyme activity. For some samples (population 1), we have evaluated nitrite (biomarker of NO bioavailability) but no correlation was observed between plasma eNOS and plasma nitrite (see below the graphics).

Specific comments

  1. How have you distinguished between PE and GH as both are HPD. Write in text in Abstract and Introduction for the clarity of readers.

We have made corrections by briefly explaining GH and PE diagnostic differences in the Abstract (lines 26-27) and Introduction (lines 74-76).

  1. Fig 1: Please show the eNOS level comparison with GH and HP with statistical analysis and GH and NP with statistical analysis (as GH have a range of above 400 pg/ml while NP is below 300 pg/ml, please check if there’s a significant difference)

We conducted this statistical analysis and have included the results in the supplementary file (Figure S1).

  1. The author should also make a figure 2 showing plasma eNOS concentration in replicative study.

This graph is presented in Figure 1c.

Reviewer 3 Report

Comments and Suggestions for Authors

In this study the authors measure plasma eNOS levels in patients from healthy and hypertensive pregnancies. In this study, eNOS levels are reduced in patients with hypertensive disorders of pregnancy in comparison to healthy pregnancy. This a somewhat novel observation, as low eNOS levels have not been described in the plasma previously. However, numerous accounts of eNOS changes in pregnancy and hypertensive disorders of pregnancy have been made over decades, and this is simply one more such measure. The following weaknesses should be addressed:

-       What is the proposed physiological significance of eNOS in the plasma? Is it an indicator of changes in eNOS elsewhere? Does it have direct significance in NO production and vasodilation? If so, what is the proposed mechanism? This is very important because eNOS protein itself is only one of many parts to the NO production equation. Additional experiments that investigate eNOS activity or NO production would be very helpful in telling a more complete story in addition to this single observation.

-       Figure 1 as presented lacks richness. One of the potential interesting features of this data could be clinical assay and biomarker/early detection utility. In order to think critically about that aspect, one would want to see the full spread of the data. Mean +/- SEM do not provide this. Presenting the data as box plots or scatter plots with confidence intervals would be more useful.

-       There is excessive use of highlighting in the text with no purpose noted.

Author Response

Reviewer #3:

Comments and Suggestions for Authors

In this study the authors measure plasma eNOS levels in patients from healthy and hypertensive pregnancies. In this study, eNOS levels are reduced in patients with hypertensive disorders of pregnancy in comparison to healthy pregnancy. This a somewhat novel observation, as low eNOS levels have not been described in the plasma previously. However, numerous accounts of eNOS changes in pregnancy and hypertensive disorders of pregnancy have been made over decades, and this is simply one more such measure. The following weaknesses should be addressed:

-       What is the proposed physiological significance of eNOS in the plasma? Is it an indicator of changes in eNOS elsewhere? Does it have direct significance in NO production and vasodilation? If so, what is the proposed mechanism? This is very important because eNOS protein itself is only one of many parts to the NO production equation. Additional experiments that investigate eNOS activity or NO production would be very helpful in telling a more complete story in addition to this single observation.

These are valuable questions, but plasma nitrite concentrations for all the samples analyzed in the primary study were not available. Although, we conducted a correlation analysis using the data available, we found no statistically significant results. The following graphs, along with their corresponding r and p values, have been provided for reference.

Figure 1 as presented lacks richness. One of the potential interesting features of this data could be clinical assay and biomarker/early detection utility. In order to think critically about that aspect, one would want to see the full spread of the data. Mean +/- SEM do not provide this. Presenting the data as box plots or scatter plots with confidence intervals would be more useful.

We have followed the proposed suggestions and revised the display of the eNOS data. To demonstrate the entire range of our data, we have amended the column bar graphs and presented the results as violin plots, which illustrate the frequency distribution of the data, alongside the median, first and third quartiles, as well as the minimum and maximum values. Also, logarithmic functions were implemented to transform concentrations into more manageable measurement scales in the graphs.

-       There is excessive use of highlighting in the text with no purpose noted.

We have now corrected this mistake.

Round 2

Reviewer 2 Report

Comments and Suggestions for Authors

The authors response is good and precise and the MS is much improved. Interestingly the authors have the data of no corelation of nitrite with eNOS. This data they can add in the MS, in supplimentary and write their interpretation in results. 

Author Response

Journal: Diseases (ISSN: 2079-9721)

Manuscript ID: diseases-2617240

Type: Article

Title: Plasma eNOS concentration in healthy pregnancy and in hypertensive disorders of pregnancy: evidence of reduced concentrations in preeclampsia from two independent studies

Julyane N. S. Kaihara, Caroline K. Minami, Maria T. S. Peraçoli, Mariana Romão-Veiga, Vanessa R. Ribeiro-Vasques, José C Peraçoli, Ana C. T. Palei, Ricardo C. Cavalli, Priscila R. Nunes, Marcelo R. Luizon, Valeria C. Sandrim

Reviewer #2:

The author’s response is good and precise and the MS is much improved. Interestingly the authors have the data of no correlation of nitrite with eNOS. This data they can add in the MS, in supplementary and write their interpretation in results. 

We appreciate the reviewer's feedback and have incorporated the correlation findings in the supplementary materials, as well as included the methodology, discussion and conclusions in the manuscript. Thank you for your valuable input.

Reviewer 3 Report

Comments and Suggestions for Authors

Authors were responsive to review.

Author Response

Thanks for review